# Incisor Occlusion Affects Profile Shape Variation in Middle-Aged Adults

**DOI:** 10.3390/jcm10040800

**Published:** 2021-02-17

**Authors:** Georgios Kanavakis, Anna-Sofia Silvola, Demetrios Halazonetis, Raija Lähdesmäki, Pertti Pirttiniemi

**Affiliations:** 1Department of Pediatric Oral Health and Orthodontics, UZB-University Center for Dental Medicine, University of Basel, Mattenstrasse 40, CH-4058 Basel, Switzerland; 2Department of Orthodontics and Dentofacial Orthopedics, Tufts University School of Dental Medicine, Boston, MA 02111, USA; 3Oral Development and Orthodontics, Research Unit of Oral Health Sciences, Medical Faculty, University of Oulu, FI-90014 Oulu, Finland; anna-sofia.silvola@oulu.fi (A.-S.S.); raija.lahdesmaki@oulu.fi (R.L.); pertti.pirttiniemi@oulu.fi (P.P.); 4Oral and Maxillofacial Department, Oulu University Hospital, Medical Research Center Oulu (MRC Oulu), FI-90014 Oulu, Finland; 5Department of Orthodontics, School of Dentistry, National and Kapodistrian University of Athens, GR-11527 Athens, Greece; dhal@dhal.com

**Keywords:** overjet, overbite, occlusion, profile shape, adults, morphometrics

## Abstract

Background: The aim of this study was to assess the effect of overjet and overbite on profile shape in middle–aged individuals. Methods: The study population comprised 1754 46-year-old individuals, members of the 1966 Northern Finland Birth Cohort. Their profile images were digitized using 48 landmarks and semi-landmarks. The subsequent landmark coordinates were then transformed to shape coordinates through Procrustes Superimposition, and final data were reduced into Principal Components (PCs) of shape. Overjet and overbite values were measured manually, during a clinical examination. A multivariate regression model was developed to evaluate the effect of overjet and overbite on profile shape. Results: The first nine PCs described more than 90% of profile shape variation in the sample and were used as the shape variables in all subsequent analyses. Overjet predicted 21.3% of profile shape in the entire sample (η^2^overjet = 0.213; *p* < 0.001), while the effect of overbite was weaker (η^2^overbite = 0.138; *p* < 0.001). In males, the equivalent effects were 22.6% for overjet and 14% for overbite, and in females, 25.5% and 13.5%, respectively. Conclusion: Incisor occlusion has a noteworthy effect on profile shape in middle-aged adults. Its impact becomes more significant taking into consideration the large variety of genetic and environmental factors affecting soft tissue profile.

## 1. Introduction

### 1.1. Background

Facial appearance is the strongest predictor of overall attractiveness and provides cues regarding health, personality, and psychosocial traits in humans [1]. An attractive face is perceived as trustworthy, competent, and intellectual, and is, thus, an important determinant of social, romantic, and professional interactions [2,3]. Sexual dimorphism, as reflected in the presence of masculine and feminine features, as well as facial averageness and symmetry, have been associated the most with facial attractiveness [1,2,3]. To a greater or lesser degree, the above factors are associated with the anatomy of individual facial features, such as the lips, eyes, and chin. The morphology of the lower facial third, in particular, contributes significantly to the variation in facial appearance among humans [4]. During adolescent growth and development, the effect of sexual hormones is largely expressed on the lower third of the face, with females exhibiting fuller lips and a less pronounced chin than males [5,6]. In adulthood, although sexual dimorphism tends to decline, facial changes continue to manifest in both sexes through flattening of the lips and an increase in nose length [7,8].

Orthodontists have historically been studying the human face as well as the underlying skeletal structures in order to observe normal variation and, most importantly, be able to predict physiologic transformation due to growth or changes caused by orthodontic treatment. There is a strong correlation between hard and soft tissue facial morphology [9] and, thus, it is expected that structural skeletal changes or positional dental changes will directly impact facial appearance. While the existing literature does confirm the influence of orthodontic treatment on soft tissues [10], the magnitude of this effect as well as its chronicity are a topic of discussion. Given that the lips are affected by orthodontic tooth movement the most, there is a lot of focus on lip responses to sagittal or vertical movements of the anterior dentition. These movements are commonly described as changes in overjet and overbite, which are defined as the sagittal/horizontal and the coronal/vertical distance between the upper and lower anterior teeth, respectively.

The amount of incisor retraction or proclination, the thickness and length of the upper lip, as well as its tonicity are the main factors linked to soft tissue treatment outcomes [11,12,13,14,15]. However, most of the current knowledge is based on data from adolescents or young adults. With the effect of aging on facial soft tissues, their relationship to the underlying teeth is expected to change. This, however, has not been studied sufficiently despite the fact that the percentage of adults who seek orthodontic treatment for esthetic reasons continues to increase. Previous reports have shown that there is an association between traditional linear and angular soft tissue measurements to overjet and overbite in mature adults [16]. Although these measurements are easier interpreted and more often used, they do not take the entire profile into consideration. Furthermore, they are largely dependent on size and therefore do not deliver accurate shape information [17].

### 1.2. Aim and Hypotheses

The aim of this study was to present a thorough assessment of the effect that overjet and overbite have on profile shape in a large, homogenous group of middle-aged adults. Our main hypothesis was that overjet and overbite explain a significant amount of profile shape variation. Secondarily, we hypothesized that profile shape alone can predict the severity of overjet and overbite in the present population.

## 2. Materials and Methods

### 2.1. Study Sample

Our sample comprised 1964 middle-aged adults, who were members of the Northern Finland Birth Cohort 1966 (NFBC1966). All of these individuals were born and have been living within a 100 km radius surrounding the city of Oulu in Northern Finland. At the age of 46, they volunteered to participate in a follow-up visit that included a series of clinical examinations. A detailed description of the NFBC1966 cohort can be found elsewhere [18]. Out of the initial population, the present study only included subjects whose clinical records were complete and of acceptable diagnostic value and who did not have any craniofacial deformities, syndromes, or a history of facial reconstructive surgery. These criteria were fulfilled by 1754 individuals (799 males and 955 females) who comprised the final study population. The research protocol was approved by the Ethical Committee of the Northern Ostrobothnia Hospital District on September 2011 (EETTMK#: 74/2011), and written consent was obtained from all volunteers prior to participation.

The clinical examination providing information regarding subjects’ occlusal features was performed according to a previously reported protocol [19], and facial photographs were acquired in a standardized manner, as described elsewhere [16]. During the clinical examination, overjet and overbite were measured in millimeters using a manual caliper at the maximum intercuspation position of the mandible. Overjet was measured as the horizontal distance between the right maxillary incisor and the labial surface of its antagonist. When the antagonist tooth was more anteriorly than the maxillary incisor (e.g., mandibular prognathism), a negative value was recorded. Overbite was measured as the vertical distance between the same teeth (vertical overlap) and was recorded as negative in cases of an anterior open bite [19]. Demographic information and clinical occlusal measurements were exported to an Excel worksheet (Microsoft Excel, Microsoft ©, Richmond WA, USA), and facial profile images were imported into Viewbox 4 software (version 4.1.0.1 BETA, dHAL Software, Kifissia, Greece) for processing and digitization.

### 2.2. Shape Analysis

To perform digitization, a template was created using the profile image of an individual who was not included in the study population. Facial profile shape was described with two curves (upper and lower facial curvature), encompassing 48 landmarks (4 fixed and 44 sliding semi-landmarks). The entire process was carried out by one research team member (GK), as previously described [20] (Appendix A). After the digitization process was completed, semi-landmarks were allowed to slide along their respective curve between the limits set by the fixed landmarks, in order to rearrange their position according to the average shape of the entire sample. This iterative process was repeated three times until the bending energy between individual samples and the average shape was minimized, and corresponding landmarks were more homogenous to each other [21,22]. Next, a General Procrustes Analysis (GPA) was performed in order to transform landmark space coordinates into shape coordinates (Appendix A), and a Principal Component Analysis (PCA) was used for data reduction and extraction of Principal Components (PCs) describing the shape variation within the population [23].

### 2.3. Statistical Analyses

The association between profile shape and occlusion was evaluated with multivariate regression models, assuming that overjet and overbite were the predictor variables and the shape PCs the dependent variables. In all analyses, shape information was provided by the first nine PCs, which described more than 90% of the variation in the sample. The decision to include the first nine PCs was based on the broken stick method [24].

An additional discriminant analysis was performed in the entire sample to investigate if profile shape alone could accurately identify the severity of overjet and overbite. The results of the discriminant analysis were cross-validated with the “leave one out” method. All statistical analyses were conducted with “ViewBox 4.1” software and IBM SPSS Statistics for Windows (Version 26.0. IBM, Armonk, NY: IBM Corp., 2020). The level of statistical significance was set at 0.05.

## 3. Results

### 3.1. Error of the Method

In order to assess the error of the method, 120 randomly selected profile images were digitized again by the same operator, after a period of two weeks, and the methodology described above was repeated. Systematic error was determined through permutation tests (100,000 permutations) to calculate the Procrustes’ distance between digitizations performed at those time points and was found to be not statistically significant (*p* = 0.915). Random error, related to the digitization process, was determined as a percentage of total variance in shape space, as explained by PC1-PC9. This resulted in a low random error of 4.6%.

### 3.2. Shape Variation

Shape variation in the present sample has been previously described in detail [20]. The first four PCs explained more than 70% of profile shape variation (PC1: 33.1%, PC2: 23.1%, PC3: 11%, and PC4: 6.7%), while the first nine PCs collectively explained more than 90% of variation. As seen in Figure 1a–d, males and females in the present sample presented a significant difference in profile shape, which was more evident in the axis of PC3 (Figure 1d). Due to this significant dimorphism in facial morphology between sexes, and because a statistically significant difference in overjet was found between sexes (Appendix A), assessments were also performed separately for males and females.

### 3.3. Profile Shape and Anterior Occlusion

For the entire sample, the multivariate regression model showed a statistically significant but weak effect of overjet and overbite on profile shape (η^2^_overjet_ = 0.213; *p* < 0.001/η^2^_overbite_ = 0.138; *p* < 0.001). The strength of the effects remained low when the association between overjet, overbite, and profile shape was assessed separately in males and females (Table 1).

In order to explore if these effects were associated with facial changes explained by specific PCs, between-subjects’ effect sizes were also assessed and are presented in Appendix A. However, they did not reveal any PCs that were more significantly related to overjet and overbite values. These individual effects on the first four PCs (explaining more than 70% variation) are also displayed in Figure 2.

In order to visualize profile shape as related to overjet and overbite, the shape information provided by the PCs was used to create profile morphings, representing extreme negative and extreme positive overjet and overbite values (Figure 3). Upon observation of these morphings, it is evident that maximal positive overjet values were associated with a convex profile, a pronounced upper lip, a deep mentolabial sulcus, and a retruded mandible. Subjects with negative overjet values presented a retrusive upper lip, a straighter profile, and a shallow labiomental sulcus. According to the amount and direction of shape variation depicted in the upper-middle morphing on Figure 3, differences in profile shape were primarily related to changes in the position of the upper lip. Extreme positive overbite values (deep bites) were associated with a reduced vertical dimension of the lower facial third and a deep mentolabial sulcus, while negative overbite (open bites) presented with an increased lower facial third and a straight profile. Shape variation, as related to overbite values, was mainly a result of lower lip projection and straightening of the mentolabial curve.

The finding of a weak but statistically significant effect of overjet and overbite on profile shape indicates a significant association but a low predictive value for these factors. In order to further assess this relationship, a discriminant analysis with cross validation was conducted to determine if profile shape could predict the severity of overjet or overbite of a subject within the sample. For the purpose of this analysis, the sample was divided into groups according to the severity of overjet and overbite, as presented in Table 2. The results show that profile shape was not able to accurately predict if an individual presented with extreme (positive or negative) values of overjet or overbite. There was a tendency of subjects being classified on the basis of their profile as having a normal overjet and overbite, confirming the weak effect of the occlusal variables on profile shape.

## 4. Discussion

This investigation assesses the amount of profile shape variation explained by overbite and overjet in a large population of middle-aged adults who all had the same exact age and ethnic identity. Due to these baseline characteristics, environmental and ethnic effects are less likely to have had an effect on the results.

The clinical importance of the study lies in that there are unique facial features seen in more mature adults, primarily a flattening of the lips and a larger nose projection [23,25], which significantly differentiate their appearance to the one of younger individuals. In recent years, there has been a steep increase in adults seeking orthodontic care, primarily to address their esthetic concerns. However, due to the aging process, the esthetic principles that are used for the treatment of adolescents and young adults mostly do not apply to middle–aged adults.

Here, we show the effect of overjet and overbite on the profile shape of middle-aged individuals. Overjet alone predicted 21.3% of profile shape variation in the entire sample, while the effect of overbite was 13.8%. These results allow us to accept our main study hypothesis. Similar findings were revealed when males and females were examined separately. Statistically these effects were significant but weak. Thus, as an isolated observation this is not in line with the notion that occlusion strongly affects facial profile, which is well-documented in growing individuals and young adults [26]. However, considering that facial profile is influenced by a large variety of factors, including the skeletal pattern and the genetically determined characteristics of the soft tissues, the present findings have a meaningful biological significance. They show that anterior occlusion, although primarily determined by tooth relationships, has a broader contribution to facial appearance. Halazonetis [9] conducted a similar study associating dentoskeletal structures to soft tissue profile shape in growing individuals and reported that the former were able to predict approximately 50% of shape variation. From a biological point of view, the dentoskeletal structures used as predictors in that study would certainly be expected to have had a larger impact on facial profile compared to overjet and overbite alone. However, overjet and overbite are also related to the overall skeletal pattern, with hyperdivergent individuals, for example, presenting greater percentages of anterior open bites. This association between skeletal pattern and incisor occlusion might partly explain their impact on facial profile. It can be speculated that if in this study we had also included more dentoskeletal variables in our analyses, we would have probably found a larger effect. However, our aim was to focus on the relationship of the anterior dentition, it being the main feature contributing to smile esthetics and to the successful outcome of orthodontic treatment.

Moreno et al. also explored profile variability in adult populations by investigating subjects with moderate to severe Class II and Class III malocclusions [27,28]. Both of these studies used cephalometric images of non-growing individuals and performed a principal component analysis to identify patterns of craniofacial morphology. In the Class II sample [27], including subjects with an overjet greater than 4 mm, they showed that 33% variation was explained by PCs related to overjet. In the Class III sample [28], including subjects with a negative overjet, approximately 34% of the phenotypic variation was explained by PCs related to incisor relationship. Both of these studies only examined cases with severe malocclusions and included a broad age spectrum of non-growing individuals, ranging between 16 and 60 years. Due to these differences, they are not directly comparable to the present study evaluating a large sample from the general population at 46 years of age. However, it is evident that the relationship of the anterior teeth described a similar percentage of phenotypic variability to the one reported here. The difference may be attributed to the fact that Moreno et al. only included cases with malocclusions.

Our results imply that it would be incorrect to undervalue the importance of obtaining optimal incisor occlusion in the treatment of older orthodontic patients. Indeed, overjet and overbite might not have a strong effect on the shape of the entire profile, but they do affect the position of the lips [11,16,29,30]. The profile morphings created in regard to overjet and overbite values of the present study population show that the largest variability in profile shape was related to the position of the upper lip (Figure 3). This finding is in agreement with the literature supporting that orthodontic repositioning of the anterior dentition has a measurable effect on the lips. Primarily in non-growing patients, where the effect of growth decreases significantly, the lips are likely the only facial structures that orthodontic treatment can impact. As presented here and in previous reports [11,16], this is particularly true for the upper lip.

In order to conduct a thorough assessment of our study aim, a discriminant analysis was also performed to reverse our initial assumption and evaluate the predictive value of profile pictures in determining anterior occlusal relationships. Interestingly, in our sample, facial profile alone was not able to predict overjet or overbite, even in extreme cases; only 10% of cases with an overjet larger than 5 mm could be correctly identified using the shape information provided by the shape PCs. This question has not been adequately addressed in the current literature. There is one study by Staudt and Kiliaridis [31] exploring the relationship between soft tissue profile and severity of Class III malocclusion in adults, showing that angular measurements on profile images may predict the underlying skeletal pattern but only in moderate to severe cases. However, their study is not comparable to the present one, because the sample only comprised individuals with a previous diagnosis of Class III malocclusion and thus was not representative of the general population. In our sample, the prevalence of negative overjet was low (64 out of 1754) and there were only a few cases with an overjet smaller than −2 mm (12 out of 64). The low predictive value, as shown here, of the entire profile shape even in severe overjet and overbite cases also implies that although there might be a significant impact of anterior occlusion on soft tissue profile, this impact is not strong enough to create a distinct phenotypic differentiation between individuals.

## 5. Conclusions

This study shows that overjet and overbite have a weak but significant impact on facial profile in middle-aged adults of Northern European descent. The upper lip displayed the largest amount of variation as related to severity of overjet. Overbite differences had a marginal effect on overall profile shape. These findings need to be interpreted within the context of our study population, which represented the general population of the particular geographic region and did not comprise individuals seeking orthodontic care. In addition, it was shown that profile shape was not able to predict anterior occlusion, and this was also evident in severe cases of excessive overjet and overbite values.

## Figures and Tables

**Figure 1 jcm-10-00800-f001:**
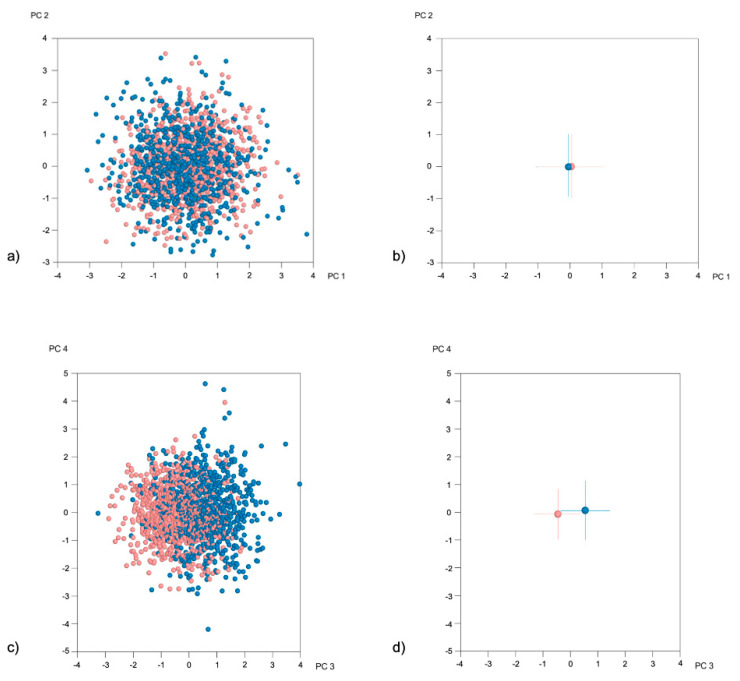
(**a**) Principal Component Analysis (PCA) graph displaying facial variation according to sex (in SD units), in the entire sample, as explained by PC1 (33.1%) and PC2 (23.1%) (Females: light red, Males: blue). (**b**) Average male (blue) and average female (light red) shape with respective SDs, as described by PC1 and PC2. (**c**) PCA graph displaying facial variation according to sex (in SD units), in the entire sample, as explained by PC3 (11%) and PC2 (6.7%) (Females: light red, Males: blue). (**d**) Average male (blue) and average female (light red) shape with respective SDs, as described by PC3 and PC4.

**Figure 2 jcm-10-00800-f002:**
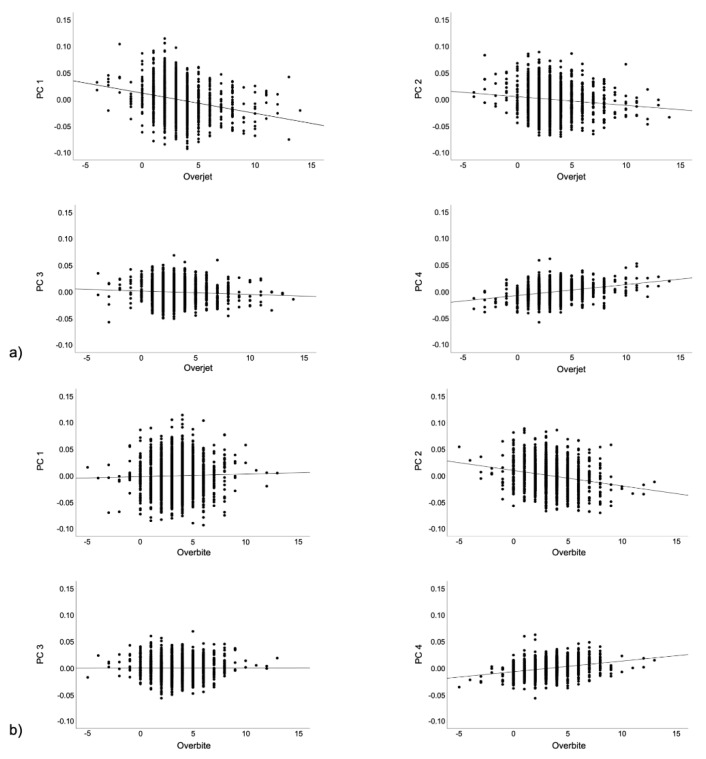
Association between overjet (**a**) and overbite (**b**) and the first primary PCs, explaining >70% of the sample variability.

**Figure 3 jcm-10-00800-f003:**
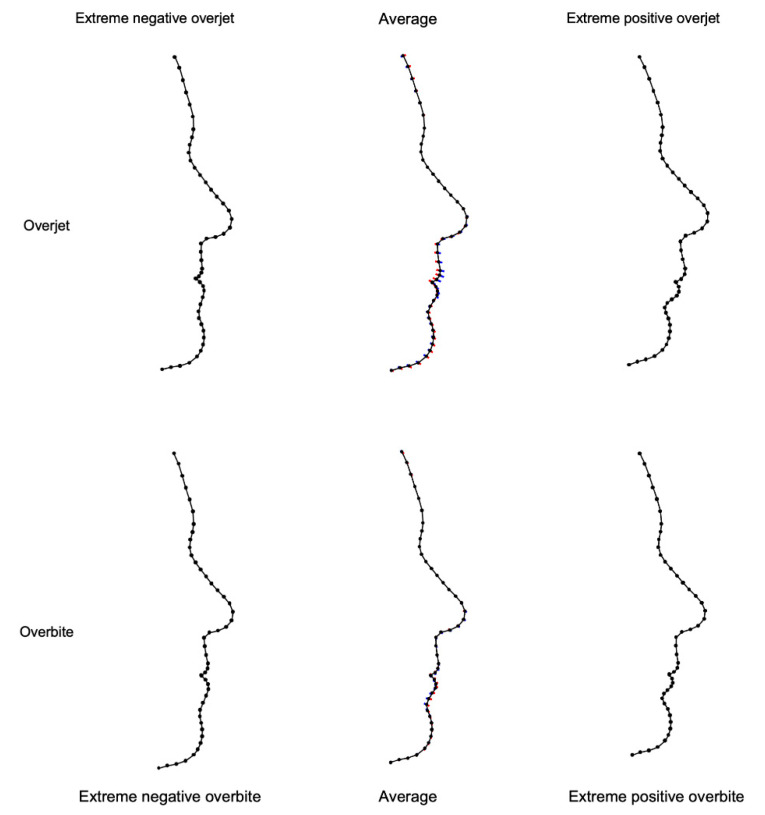
Profile morphings according to overjet (top row) and overbite (bottom row). The middle column depicts the average facial shape of the entire sample, as well as the amount and direction of variation at each profile landmark (blue: positive, red: negative), as explained by overjet (top row) and overbite (bottom row) values. On each row, the images left and right of the average present profile shapes of extreme overjet (top) and overbite (bottom) values.

**Table 1 jcm-10-00800-t001:** Effect of overjet and overbite on profile shape, as explained by PC1-PC9.

	N	Predictor	η^2^	*p*-Value
Entire Sample	1754	Overjet	0.213	<0.001
		Overbite	0.138	
Males	799	Overjet	0.226	
		Overbite	0.140	
Females	955	Overjet	0.255	
		Overbite	0.135	

**Table 2 jcm-10-00800-t002:** Discriminant analysis of the predictive value of profile shape on severity of overjet and overbite.

Predictors			N	Predicted Group			Predictive Percentage
				Negative	Normal	Excessive	
**PC1—PC9**	Overjet groups	Negative (≤0 mm)	64	3	61	0	4.7%
		Normal (1–5 mm)	1530	4	1509	17	98.6%
		Excessive (>5 mm)	160	0	144	16	10.0%

	Overbite groups	Negative (≤0 mm)	92	1	91	0	1.1%
		Normal (1–5 mm)	1454	2	1440	12	99.0%
		Excessive (>5 mm)	208	0	193	15	7.2%

## Data Availability

NFBC data are available from the University of Oulu, Infrastructure for Population Studies. Permission to use the data can be applied for research purposes via electronic material request portal. In the use of data, we follow the EU general data protection regulation (679/2016) and Finnish Data Protection Act. The use of personal data is based on cohort participant’s written informed consent at his or her latest follow–up study, which may cause limitations to its use. Please contact NFBC project center (NFBCprojectcenter@oulu.fi) and visit the cohort website (www.oulu.fi/nfbc (accessed on 16 February 2021)) for more information.

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
