# Peer review of "Incisor Occlusion Affects Profile Shape Variation in Middle-Aged Adults"

_jcm, 2021, doi:10.3390/jcm10040800_

Round 1

Reviewer 1 Report

Well written manuscript. The reviewer just has few minor comments:

  1. The terms overbite and overjet are the main elements of the manuscript. However, even though it was mentioned in the author's other article, the description of the measurement wasn't clear in this manuscript. Please consider adding more detailed information to this measurement process.
  2. The reader of this journal may include non-dentist. For those non-dental readers, a short definition of overbite/overjet might be helpful to be added in the Introduction section.

Author Response

We would like to thank the reviewer for his constructive comments that have helped us improve our original submission. Please see our detailed responses below.

Comments and Suggestions for Authors

Well written manuscript. The reviewer just has few minor comments:

  1. The terms overbite and overjet are the main elements of the manuscript. However, even though it was mentioned in the author's other article, the description of the measurement wasn't clear in this manuscript. Please consider adding more detailed information to this measurement process.

Authors’ response: The reviewer makes a valid point. We have added some information in the description of the measurement process to facilitate better understanding for the readership. New text has been added in the revised manuscript between lines 105-112 and reads as follows: During the clinical examination, overjet and overbite were measured in millimeters using a manual caliper at the maximum intercuspation position of the mandible. Overjet was measured as the horizontal distance between the right maxillary incisor and the labial surface of its antagonist. When the antagonist tooth was more anteriorly than the maxillary incisor (e.g. mandibular prognathism), a negative value was recorded. Overbite was measured as the vertical distance between the same teeth (vertical overlap) and was recorded as negative in cases of an anterior openbite [19].  

  1. The reader of this journal may include non-dentist. For those non-dental readers, a short definition of overbite/overjet might be helpful to be added in the Introduction section.

Authors’ response: The reviewer is correct. This information was missing in the initial manuscript. There has been new text added in the revised version of the manuscript, between lines 68-70. This part of the introduction now reads: These movements are commonly described as changes in overjet and overbite, which are defined as the sagittal/horizontal and the coronal/vertical distance between the upper and lower anterior teeth, respectively.

Reviewer 2 Report

Dear Authors, thank you for submitting your paper.

The aim of the present study was to assess the effect of overjet and overbite on pro- 25 file shape in middle-aged individuals.

I congratulate the authors for this very relevant research, which will add to the dental field.

The study is interesting and It appears well structured, correctly carried out and written without logical or factual errors

Please introduce in the Introduction section a subparagraph for the aim of the study.

Methodological aspects are deeply cleared in the manuscript. 

The topic is in line with the journal aim.

-Could you provide an explanatory Figure for the Shape analysis

-Data reported in the Methods section are appropriate and precisely described.

-Results are reported clearly and adequately supported by Tables.

-I suggest to the Authors to improve their reference list adding the following two manuscript:

https://doi.org/10.3390/nu12123688

https://doi.org/10.3390/ijerph17239104

The Conclusions are correctly stated and supported by the findings obtained from the present study.This study shows that overjet and overbite have a weak, but significant impact on facial  profile in middle-aged adults of Northern European descent. Moreover, The present research should be performed in future to asses the same parameters in subjects from other geographic regions.

According to this Reviewer’s consideration, novelty and quality of the paper, publication of the present manuscript is recommended.

Author Response

We thank the reviewer for his valuable suggestions and comments. They have helped us in improving our initial submission. We would also like to thank the reviewer for suggesting publication of this manuscript.
Please see our detailed responses below.

Comments and Suggestions for Authors

Dear Authors, thank you for submitting your paper.

The aim of the present study was to assess the effect of overjet and overbite on profile shape in middle-aged individuals.

I congratulate the authors for this very relevant research, which will add to the dental field.

The study is interesting and It appears well structured, correctly carried out and written without logical or factual errors

Authors’ response: We thank the reviewer for his comments.

Please introduce in the Introduction section a subparagraph for the aim of the study.

Authors’ response: The introduction section of the revised manuscript has been divided into the following subsections: 1.1 Background and 1.2. Aim and Hypotheses.

Methodological aspects are deeply cleared in the manuscript. 

The topic is in line with the journal aim.

-Could you provide an explanatory Figure for the Shape analysis

Authors’ response: Two supplementary figures have been added to the revised manuscript in order to facilitate understanding of the digitization process and the Procrustes Superimposition. These are named Figure S1 (line 121) and Figure S2 (line 128) and are included in the Supplementary Material of the revised submission.

-Data reported in the Methods section are appropriate and precisely described.

Authors’ response: We thank the reviewer for the comment.

-Results are reported clearly and adequately supported by Tables.

Authors’ response: We thank the reviewer for the comment.

-I suggest to the Authors to improve their reference list adding the following two manuscript:

https://doi.org/10.3390/nu12123688

https://doi.org/10.3390/ijerph17239104 

Authors’ response: We thank the reviewer for the suggestions. We have read both articles and have found them very interesting, nevertheless we do not think they fall within the scope of the present manuscript. However, we will certainly consider them for future publications.

The Conclusions are correctly stated and supported by the findings obtained from the present study.This study shows that overjet and overbite have a weak, but significant impact on facial  profile in middle-aged adults of Northern European descent. Moreover, The present research should be performed in future to asses the same parameters in subjects from other geographic regions.

According to this Reviewer’s consideration, novelty and quality of the paper, publication of the present manuscript is recommended.

Authors’ response: We thank the reviewer for the constructive comments and suggestions, and for his decision to recommend publication of our work.